# Deep learning insights into the architecture of the mammalian egg-sperm fusion synapse

**Arne Elofsson[1]\*, Ling Han[2], Enrica Bianchi[3†], Gavin J Wright[3], Luca Jovine[2]\***

[1]Science for Life Laboratory and Department of Biochemistry and Biophysics, Stockholm University, Solna, Sweden; [2]Department of Biosciences and Nutrition, Karolinska Institutet, Huddinge, Sweden; [3]Department of Biology, Hull York Medical School, York Biomedical Research Institute, University of York, York, United Kingdom

**\*For correspondence:**
arne@bioinfo.se (AE);
luca.jovine@ki.se (LJ)

**Present address:** †Department of Biomedicine and Prevention, University of Rome Tor Vergata, Rome, Italy

**Abstract** A crucial event in sexual reproduction is when haploid sperm and egg fuse to form a new diploid organism at fertilization. In mammals, direct interaction between egg JUNO and sperm IZUMO1 mediates gamete membrane adhesion, yet their role in fusion remains enigmatic. We used AlphaFold to predict the structure of other extracellular proteins essential for fertilization to determine if they could form a complex that may mediate fusion. We first identified TMEM81, whose gene is expressed by mouse and human spermatids, as a protein having structural homologies with both IZUMO1 and another sperm molecule essential for gamete fusion, SPACA6. Using a set of proteins known to be important for fertilization and TMEM81, we then systematically searched for predicted binary interactions using an unguided approach and identified a pentameric complex involving sperm IZUMO1, SPACA6, TMEM81 and egg JUNO, CD9. This complex is structurally consistent with both the expected topology on opposing gamete membranes and the location of predicted N-glycans not modeled by AlphaFold-Multimer, suggesting that its components could organize into a synapse-like assembly at the point of fusion. Finally, the structural modeling approach described here could be more generally useful to gain insights into transient protein complexes difficult to detect experimentally.

## eLife assessment

This study offers **valuable** insights into the structural architecture of the mammalian egg-sperm fusion synapse, shedding light on the role of specific proteins in fertilization. The significance of the findings lies in the potential identification of a pentameric complex involved in gamete fusion by AlphaFold Multimer. The strength of evidence for the approach/methodology is **solid**, while the experimental validation is **incomplete** in supporting these interactions. This work will be of interest to biomedical researchers working on fertility and reproductive health.

## Introduction

By merging the plasma membranes of egg and sperm and combining genetic material to initiate the development of a new individual, gamete fusion is the culmination of fertilization and a fundamental event in the life cycle of sexually reproducing species. Significant advances during the last twenty years have started to unravel the molecular basis of this phenomenon by identifying proteins essential for this process in organisms ranging from unicellular algae to mammals (*Clark, 2018*;

*Deneke and Pauli, 2021*). In particular, recognition between egg glycosylphosphatidylinositol-anchored protein JUNO and sperm type I transmembrane protein IZUMO1 was found to be essential for the fusion of mouse gametes by mediating the juxtaposition of their plasma membranes (*Bianchi et al., 2014*; *Inoue et al., 2005*). In agreement with such a docking function, structural studies showed that, although the architecture of the ectodomain of IZUMO1 is reminiscent of *Plasmodium* invasion proteins, neither molecule resembles known fusogens (*Aydin et al., 2016*; *Han et al., 2016*; *Kato et al., 2016*; *Nishimura et al., 2016*; *Ohto et al., 2016*); at the same time, mouse IZUMO1 was recently reported to have fusogenic activity in vitro (*Brukman et al., 2023*), but whether this reflects a comparable function in vivo remains to be determined (*Bianchi and Wright, 2023*).

Despite the importance of the JUNO/IZUMO1 interaction for gamete fusion, gene ablation experiments in the mouse have identified several other egg and sperm molecules essential for this process. On the female side, these include two phylogenetically close tetraspanin membrane proteins, CD9 and CD81 (*Miyado et al., 2000*; *Kaji et al., 2000*; *Miller et al., 2000*; *Rubinstein et al., 2006*). CD9 concentrates to the gamete adhesion area concomitantly with IZUMO1 (*Chalbi et al., 2014*) and is thought to facilitate fusion by reshaping the oocyte's plasma membrane (*Jégou et al., 2011*; *Umeda et al., 2020*). CD81 is 44%-sequence identical to CD9 and can partially rescue the infertility of CD9-deficient mouse eggs (*Kaji et al., 2002*; *Ohnami et al., 2012*). On the male side, several surface-expressed molecules are required for mouse gamete fusion in addition to IZUMO1. These include sperm acrosome membrane-associated protein 6 (SPACA6; *Barbaux et al., 2020*; *Lamas-Toranzo et al., 2020*; *Lorenzetti et al., 2014*; *Noda et al., 2020*) and transmembrane protein 95 (TMEM95; *Lamas-Toranzo et al., 2020*), both of which are type I-transmembrane proteins with an IZUMO1-like ectodomain structure (*Lamas-Toranzo et al., 2020*; *Nishimura et al., 2016*; *Vance et al., 2022*). Sperm dendrocyte expressed seven transmembrane protein domain-containing proteins 1 and 2 (DCST1/2), which interact with each other (*Noda et al., 2022*) and are orthologues of molecules essential for fusion in worm (SPE-49/42) (*Kroft et al., 2005*; *Wilson et al., 2018*) and fly (SNEAKY/DCST2) (*Wilson et al., 2006*), are required for fertility not only in the mouse but also in fish (*Inoue et al., 2021*; *Noda et al., 2022*). Finally, two other molecules necessary for mouse gamete fusion are fertilization influencing membrane protein (FIMP), the transmembrane domain-containing isoform of 4930451I11RIK (*Fujihara et al., 2020*), and sperm-oocyte fusion required 1 (SOF1; *Noda et al., 2020*). In addition to this gene knockout-derived information, there is biochemical evidence that IZUMO1 is part of rodent sperm multiprotein complexes that include structurally related molecules IZUMO2-4 (*Ellerman et al., 2009*). More recently, egg Fc receptor-like 3 (FCRL3/MAIA) was also suggested to be involved in human gamete adhesion and fusion by replacing JUNO as an IZUMO1-binding partner (*Vondrakova et al., 2022*), although others have not confirmed this (*Bianchi et al., 2024*).

The relatively large number of proteins that these studies collectively identified as required for mammalian egg-sperm fusion, together with the lack of conclusive evidence supporting a direct role of the JUNO/IZUMO1 complex in the fusion process itself, suggest that — in line with the concept of fertilization synapse (*Krauchunas et al., 2016*) — a larger macromolecular complex may orchestrate fusion. However, perhaps because such an assembly exists only transiently due to the need to prevent polyspermy, the identification of additional protein-protein interactions between the aforementioned factors has frustrated independent efforts by multiple laboratories.

Here, we show that, despite the clear centrality of the JUNO/IZUMO1 interaction, its mouse components have such a low affinity that, unlike their human homologs, they cannot be purified as a stable complex. Because the biochemical identification of other egg/sperm fusion factor complexes may be hindered by the fact that their binary affinities also vary significantly among different species, we attack the problem by taking advantage of the momentous advances in protein complex structure prediction using AlphaFold-Multimer (*Burke et al., 2023*; *Evans et al., 2021*). The rationale for using this approach lies in the fact that the availability of a significant number of sequences for the proteins of interest not only allows AlphaFold to predict possible complexes thereof highly accurately (*Jumper et al., 2021*; *Lee et al., 2023*; *Mirdita et al., 2022*), but also makes it largely insensitive to the species-specific affinity of a given protein-protein interaction.

Consistent with these considerations, the analysis of AlphaFold-Multimer predictions supports the suggestion that JUNO and IZUMO1 are part of a complex that includes additional fusion factors.

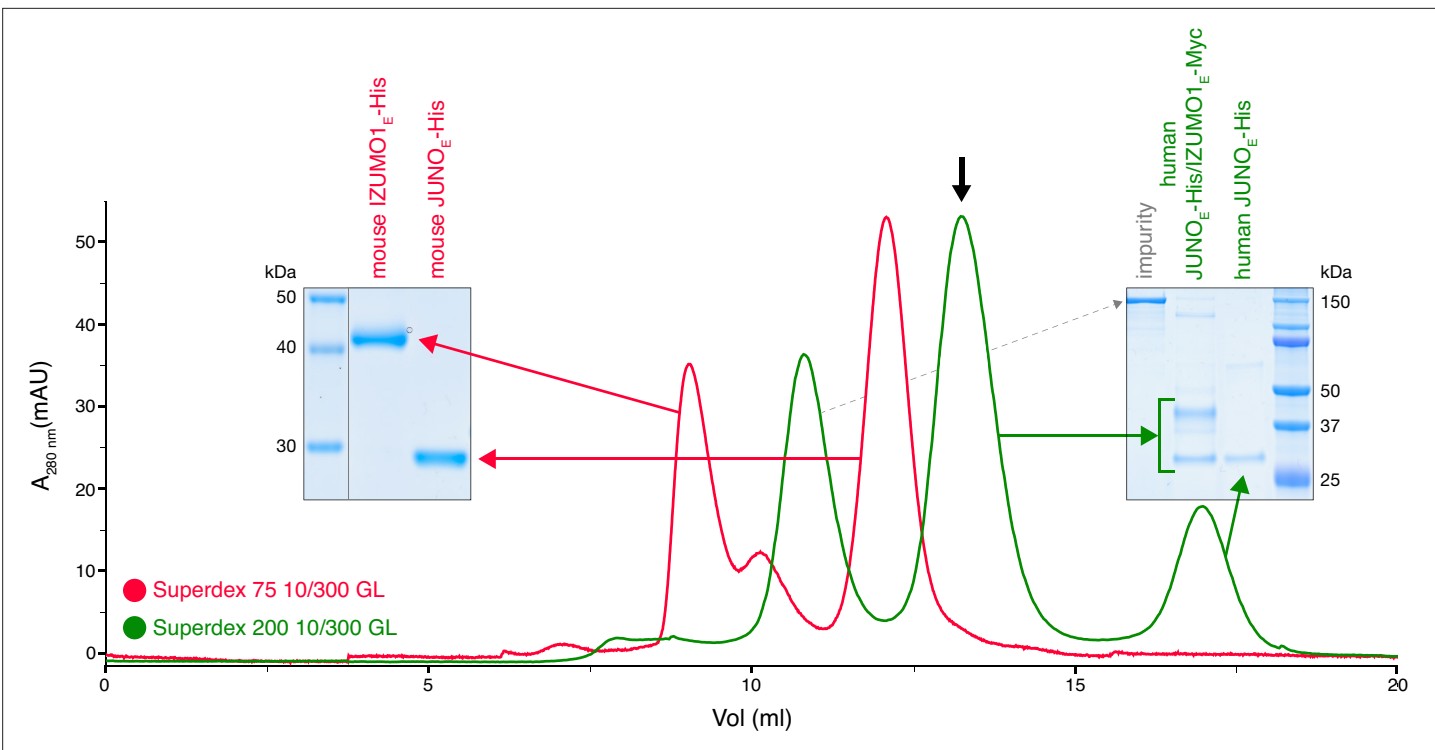

**Figure 1.** Human but not mouse JUNO and IZUMO1 ectodomains form a stable complex in solution. The SEC elution profile of immobilized metal affinity chromatography (IMAC)-purified human $JUNO_E$-His/$IZUMO1_E$-Myc (green trace) shows a major peak that contains both proteins (black arrow), as well as a peak corresponding to unbound $JUNO_E$-His (SDS-PAGE analysis on the right). On the contrary, IMAC-purified mouse $JUNO_E$-His and mouse $IZUMO1_E$-His elute separately on gel filtration (red trace and SDS-PAGE analysis on the left). See also *Figure 1—figure supplement 1*.

The online version of this article includes the following source data and figure supplement(s) for figure 1:

**Source data 1.** Uncropped gel scans for *Figure 1*.

**Figure supplement 1.** Immunoblot analysis of the SEC peak corresponding to the human JUNO/IZUMO1 ectodomain complex.

**Figure supplement 1—source data 1.** Uncropped blot scans for *Figure 1—figure supplement 1*.

## Results

### Mouse JUNO and IZUMO1 do not form a biochemically stable complex

Whereas mammalian cell-expressed human JUNO and IZUMO1 ectodomains form a stable complex ($JUNO_E$/$IZUMO1_E$) that can be detected by size-exclusion chromatography (SEC), their murine homologs do not (*Figure 1* and *Figure 1—figure supplement 1*). This is consistent with the low affinity of the interaction between the mouse proteins, whose 0.6–12 µM $K_D$ is significantly higher than the ~50–90 nM $K_D$ reported for the human $JUNO_E$/$IZUMO1_E$ complex expressed in insect cells (*Aydin et al., 2016*; *Bianchi et al., 2014*; *Nishimura et al., 2016*; *Ohto et al., 2016*). Notably, the $K_D$ of wild-type mouse $JUNO_E$/$IZUMO1_E$ is also higher than the 360 nM $K_D$ of the complex between human $IZUMO1_E$ and $JUNO_E$ W62A (*Aydin et al., 2016*; *Ohto et al., 2016*). The latter bears an interface mutation whose introduction into mouse JUNO abolishes its ability to rescue the sperm-fusion impairment of *Juno* null eggs, as well as halves its ability to support sperm binding to JUNO-expressing human embryonic kidney 293T (HEK293T) cells (*Kato et al., 2016*). The low affinity of mouse $JUNO_E$/$IZUMO1_E$ could, in principle, be partially compensated by the avidity resulting from a high local concentration of receptors at the egg/sperm contact point. At the same time, consistent with the considerations made above, the binary interaction between JUNO and IZUMO1 may be stabilized within the context of a larger macromolecular complex.

## AlphaFold-Multimer produces high-confidence predictions for both mouse and human JUNO/IZUMO1 ectodomain complexes

To assess whether the significant difference in affinity between the mouse and human complexes was reflected by the confidence of the corresponding AlphaFold-Multimer predictions, we compared the output of AlphaFold-Multimer runs performed without using templates. This computational experiment showed that AlphaFold-Multimer not only generates a high-confidence model of human $JUNO_E$/$IZUMO1_E$ that accurately reproduces the corresponding crystal structure but also yields a model of mouse $JUNO_E$/$IZUMO1_E$ of comparable confidence (*Figure 2*). This is consistent with the expectation that, as long as a significant number of sequences can be aligned to those of a protein complex of interest and the interaction is evolutionarily conserved, the quality of the AlphaFold-Multimer predictions for this complex is not negatively affected by the low affinity that its components may have in a subset of species.

## TMEM81 is a structural homolog of IZUMO1 and SPACA6

Considering that IZUMO1-4, SPACA6, and TMEM95 are part of a distinct superfamily of extracellular proteins implicated in gamete fusion (*Lamas-Toranzo et al., 2020*; *Nishimura et al., 2016*; *Vance et al., 2022*), we used Foldseek (*van Kempen et al., 2024*) to scan the AlphaFold/Swiss-Prot database for further proteins of similar structure. Despite insignificant sequence identities (16–27%), this search also identified transmembrane protein 81 (TMEM81) as a clear structural homolog of the conserved immunoglobulin (Ig)-like domain of IZUMO1 and SPACA6 (E-values 1.40e-8–1.39e-6; *Figure 3A and B*). The TMEM81 hit was confirmed by the result of a search of the PDB database, carried out by generating an AlphaFold model of the protein's ectodomain residues A31-N184 (average pLDDT 93.8) and using it as input for Dali (*Holm, 2020*), which matched it to the crystal structure of human IZUMO1 (PDB 5JK9; *Ohto et al., 2016*) with a Z-score of 13.0 (significantly above the Z-score threshold of 8, which indicates very good structural superpositions *Holm, 2020*). Notably, TMEM81 is conserved in vertebrates (*NCBI, 2022*), and its gene is expressed in both mouse and human spermatids (*Jung et al., 2019*; *Uhlén et al., 2015*; *Yue et al., 2014*). Like IZUMO1-3, SPACA6, and TMEM95, TMEM81 is predicted to be a type I transmembrane protein with a large extracellular domain; moreover, it was previously anonymously suggested to be a β-sheet-rich molecule that may be structurally related to IZUMO1 (*Wikipedia, 2020*). Accordingly, the characteristic four-helix bundle (4HB) of IZUMO1 and SPACA6 is replaced by a three-stranded β-sheet in the AlphaFold model of TMEM81; however, the positioning of two invariant disulfide bonds that orient these highly different elements relative to the conserved Ig-like domain is remarkably similar in the three molecules (*Figure 3C*).

## Prediction of interactions between human proteins associated with gamete fusion

To infer whether a larger macromolecular complex may be involved in gamete fusion without introducing a large number of possible false positives (as observed in attempts to perform a large-scale screening), we used AlphaFold-Multimer in template-free mode to examine all pairwise interactions of the human homologs of the 4 egg and 11 sperm proteins mentioned above, including TMEM81. Since we also considered the possibility that each of these 15 different molecules may also homodimerize, this amounted to a total of 120 unique combinations. Analysis of the corresponding predictions revealed a cluster of 7 possible interactions centered around IZUMO1, 5 of which were direct (JUNO, CD9, CD81, SPACA6, TMEM81) and 2 indirect (IZUMO4 (via JUNO), SOF1 (via SPACA6)). In addition, we detected isolated homodimeric interactions for IZUMO4 and DCST1, as well as heterodimeric interactions for IZUMO2/IZUMO3, TMEM95/FIMP and DCST1/DCST2 (*Figure 4* and *Supplementary file 1*). Notably, the ~260 Å-long mace-shaped heterodimeric assembly predicted for DCST1/DCST2 is consistent with experimental evidence for interaction between the two proteins (*Noda et al., 2022*).

To assess the relative contribution of the components of the 7-interaction cluster, we used AlphaFold-Multimer to model the corresponding 8-protein complex. Analysis of the resulting predictions (*Figure 5A* and *Figure 5—figure supplement 1A*), as well as the predictions of the binary complexes IZUMO1/CD9 (*Figure 5—figure supplement 1B*) or IZUMO1/CD81 (*Figure 5—figure supplement 1C*), suggest that the two egg tetraspanins are interchangeable because they are predicted to bind to the same region of IZUMO1; moreover, in agreement with the observation that mouse fertility depends more on CD9 than CD81 (*Kaji et al., 2002*; *Kaji et al., 2000*; *Miller et al.,*

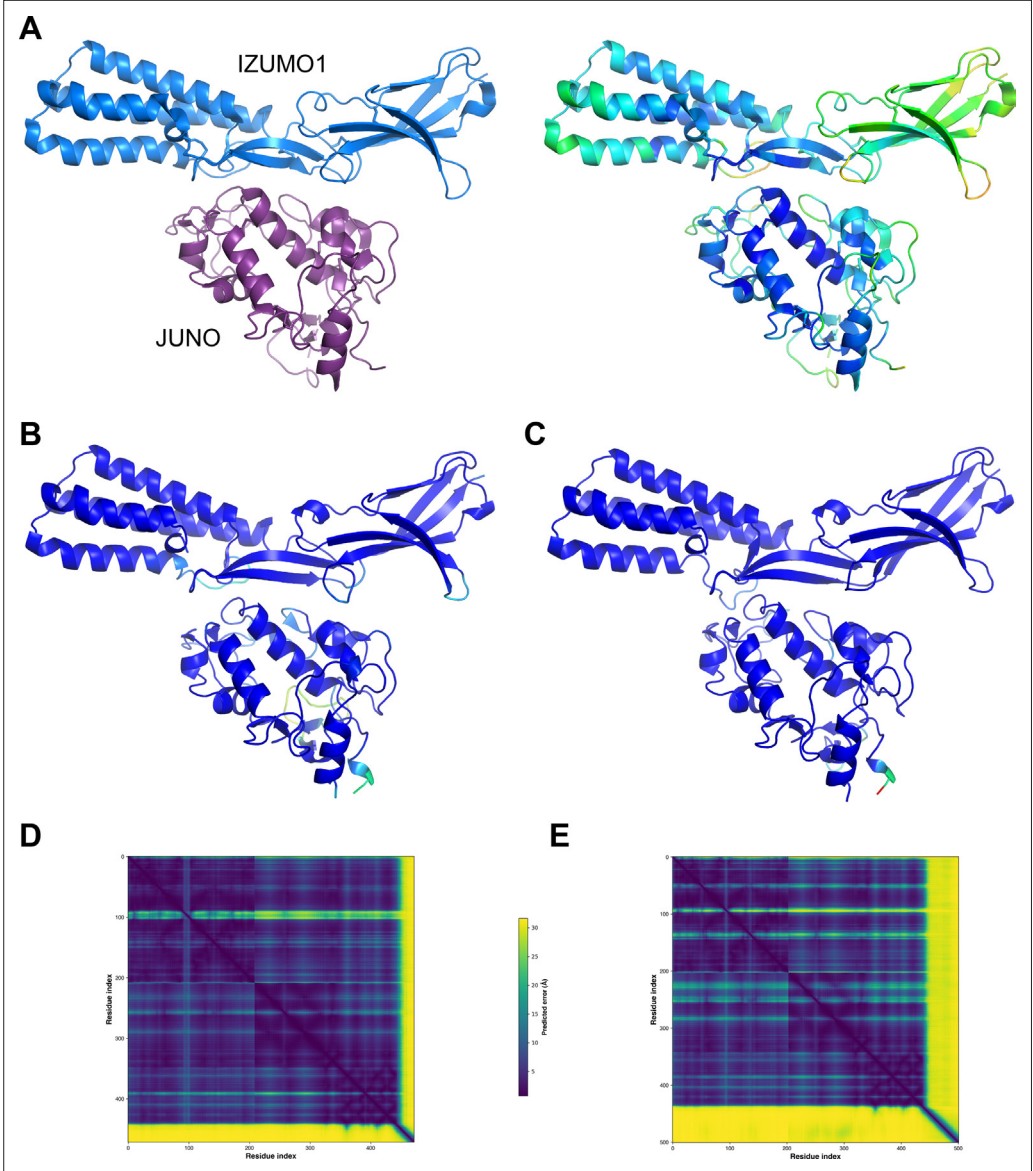

**Figure 2.** Mouse JUNO-IZUMO1 complex structure prediction. (**A**) The crystal structure of the human JUNO$_E$/ IZUMO1$_E$ complex (PDB 5F4E; *Aydin et al., 2016*), shown in cartoon representation and colored by chain (left) or by B-factor (right). (**B**) AlphaFold-Multimer template-free prediction of the structure of the human JUNO$_E$/IZUMO1$_E$ complex. The top-ranked model has a ranking confidence (rc = 0.8*predicted interface Template Modeling score (ipTM) +0.2*predicted Template Modeling score (pTM)) of 0.87, and an average root mean square deviation (RMSD) from PDB 5F4E of 2.34 Å over 437 Cα (0.88 Å over 380 Cα after outlier rejection). Only the residues that match those resolved in the crystal structure are shown; the model is colored by prediction confidence from blue to red, according to a 100-(per-residue confidence (predicted local distance difference test, pLDDT) *Jumper et al., 2021*) scale that ranges from 0 (blue; maximum confidence) to 100 (red; minimum confidence), respectively. (**C**) AlphaFold-Multimer top-ranked template-free prediction of the structure of the mouse JUNO$_E$/IZUMO1$_E$ complex (rc = 0.85; RMSD vs. 5F4E=2.53 Å over 435 Cα (1.73 Å over 389 Cα after outlier rejection)), depicted and colored as in panel B. (**D**) Predicted Aligned Error (PAE) plot for the human complex model shown in panel B. Residue indexes refer to the sequence of JUNO (amino acids G20-S228) followed by that of IZUMO1 (amino acids C22-Q284). The high PAE regions correspond to loop 2 of JUNO (residues V110-G123) and the C-terminal tail of IZUMO1$_E$ (residues K255-Q284), both of which have low pLDDT scores and are far away from the interface between the two proteins. (**E**) PAE plot of the mouse complex shown in panel C, with residue indexes referring to JUNO (amino acids G20-G222) followed by IZUMO1 (amino acids C22-R319).

The online version of this article includes the following source data for figure 2:

**Source data 1.** Input and output files for the AlphaFold-Multimer predictions shown in panels B-E.

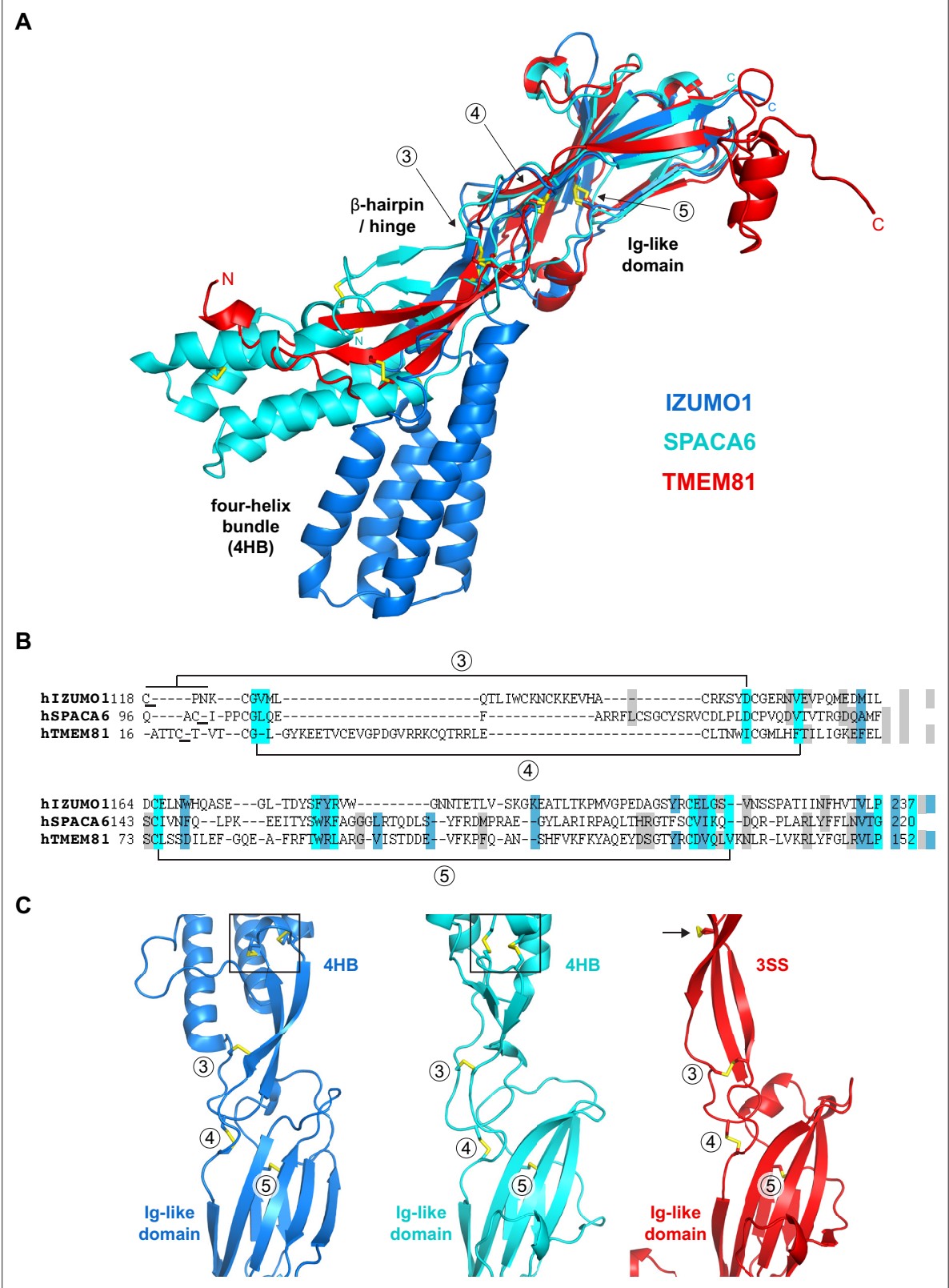

**Figure 3.** Structural homology between IZUMO1, SPACA6 and TMEM81. (**A**) Structural superposition of the ectodomains of human IZUMO1 (residues C22-K255 of PDB 5JK9 chain A; *Aydin et al., 2016*), human SPACA6 (residues C27-G246 of PDB 7TA2; *Vance et al., 2022*) and an AlphaFold model of the ectodomain of human TMEM81 (corresponding to residues I31-P218 of UniProt entry Q6P7N7, extracted from https://alphafold.ebi.ac.uk/files/AF-Q6P7N7-F1-model_v4.pdb). The three different regions of IZUMO1 and SPACA6 are indicated in black. Disulfide bonds are shown as yellow sticks,

*Figure 3 continued on next page*

*Figure 3 continued*

with arrows indicating disulfides 3–5 of IZUMO1 that are conserved in both SPACA6 and TMEM81. N- and C-termini are marked. (**B**) Structure-based alignment of the sequence regions includes conserved disulfides 3 and 4, followed by the Ig-like domain harboring conserved disulfide 5. (**C**) Partial grid view of the superposition shown in panel A, centered around the junction between the three molecules' variable (top) and conserved (bottom) domains. Note the strikingly similar relative arrangement of invariant disulfides 3, 4, and 5, and how an additional disulfide within the three-stranded sheet (3SS) of TMEM81 (black arrow) roughly matches the position of the double CXXC motifs of IZUMO1 and SPACA6 (black boxes).

*2000*; *Miyado et al., 2000*; *Ohnami et al., 2012*; *Rubinstein et al., 2006*), IZUMO1 consistently interacts with the former when modeled together with both tetraspanins. The 8-protein complex predictions also indicate that IZUMO4 does not interact with the rest of the assembly (*Figure 5A* and *Figure 5—figure supplement 1A*), consistent with the observation that its predicted binary interaction with JUNO (*Figure 4*) is incompatible with the JUNO/IZUMO1 interface (*Figure 5—figure*

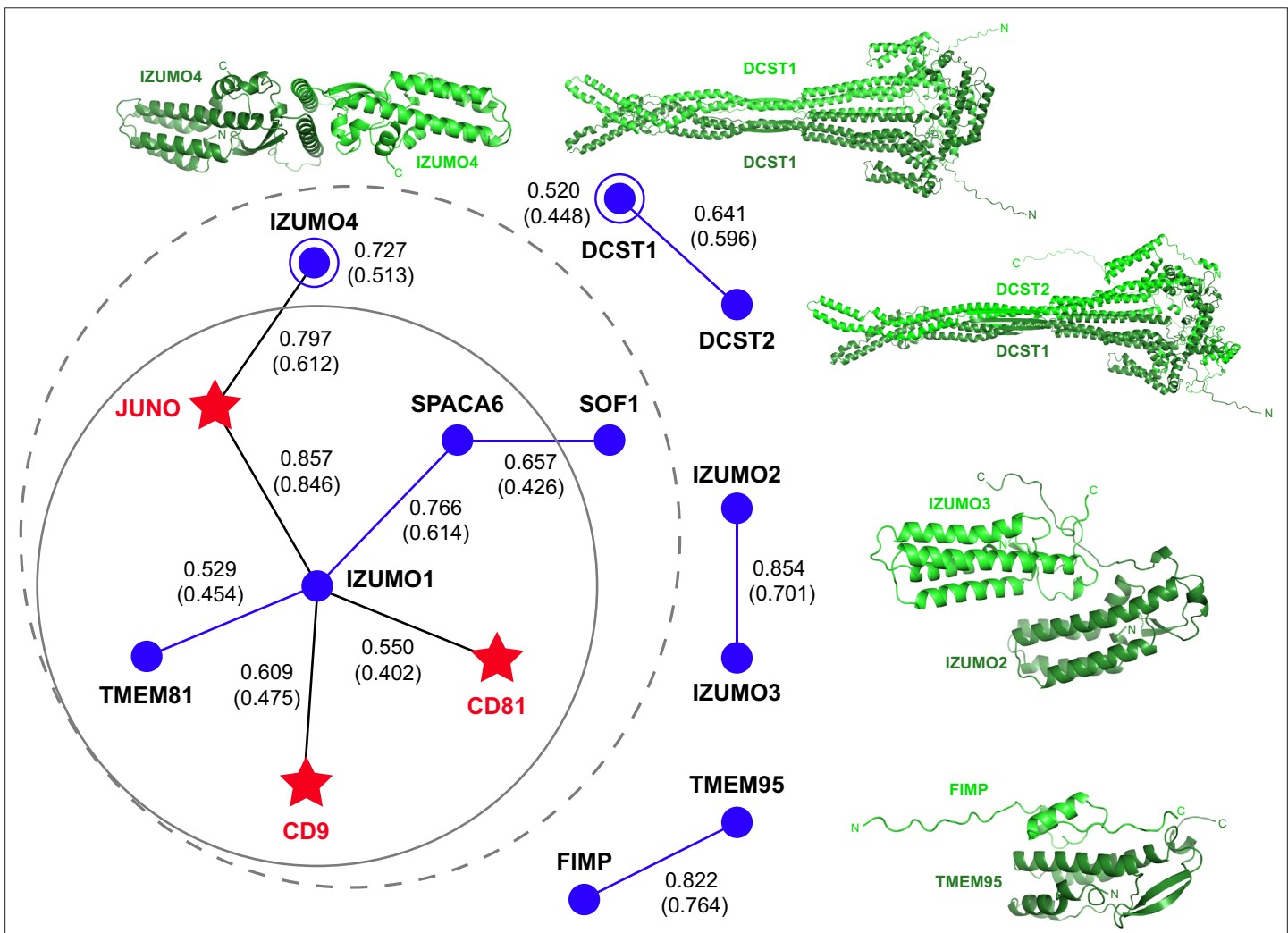

**Figure 4.** AlphaFold-Multimer prediction of interactions between fusion-associated human gamete proteins. Egg and sperm proteins are indicated by red star and blue circle symbols, respectively. Interactions between egg and sperm proteins are shown as black lines connecting the respective symbols; homomeric and heteromeric interactions between sperm proteins are depicted as blue lines and open circles, respectively. For every interaction, the top-ranking model rc is reported, with the corresponding mean rc in parenthesis (for additional metrics, see *Supplementary file 1*). The gray dashed circle indicates a network of seven interactions, identified using a mean rc cutoff of 0.4; the inner continuous circle highlights the five interactions within the network that involve sperm IZUMO1. Top-ranked predictions for the isolated binary interactions of other sperm subunits are shown in cartoon representation, with the two moieties of each complex colored dark and light green and the N- and C-termini of each chain indicated when possible.

The online version of this article includes the following source data for figure 4:

**Source data 1.** Full metrics file and PDB files for all displayed AlphaFold-Multimer predictions.

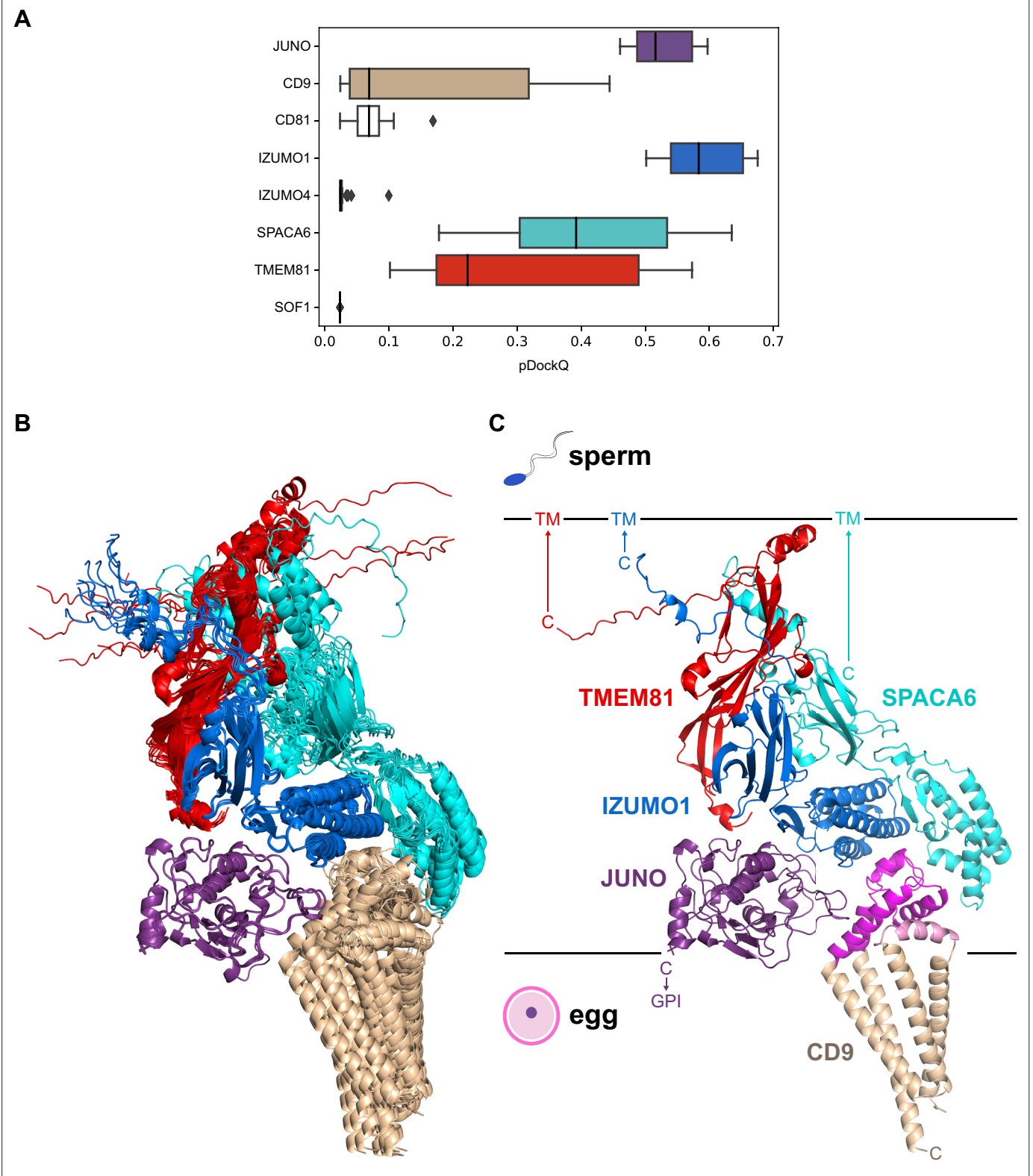

**Figure 5.** A predicted five-subunit complex at the egg/sperm plasma membrane interface. (**A**) pDockQ analysis of 25 AlphaFold-Multimer predictions for a complex consisting of the 8 proteins enclosed by the dashed gray circle in *Figure 4*. The pDockQ score for each component of every prediction was calculated with respect to the rest of the corresponding complex, and the 25 scores for each chain were then plotted as a box plot. (**B**) Superposition of the ten top-ranked AlphaFold predictions for a five-subunit complex consisting of egg CD9 and the ectodomains of egg JUNO

*Figure 5 continued on next page*

*Figure 5 continued*

and sperm IZUMO1, SPACA6 and TMEM81 (mean rc = 0.67, mean ipTM = 0.66). Proteins are shown in cartoon representation and colored by chain according to panel A. (**C**) Top-ranked model from the ensemble in panel B (rc = 0.74, ipTM = 0.73). Subunits are colored as in the previous panels, except for CD9 whose short extracellular loop (SEL) and long extracellular loop (LEL) are highlighted in pink and magenta, respectively. Ectodomain (JUNO, IZUMO1, SPACA6, TMEM81) or protein (CD9) C-termini are marked, with horizontal lines representing the approximate surfaces of the gamete plasma membranes. TM, transmembrane domain; GPI, glycosylphosphatidylinositol anchor.

The online version of this article includes the following source data and figure supplement(s) for figure 5:

**Source data 1.** Box plot data and input and output files for the AlphaFold-Multimer predictions shown in panels B and C.

**Figure supplement 1.** Modeling of the 8-protein network and binary subcomplexes thereof.

**Figure supplement 1—source data 1.** PDB files for all displayed AlphaFold-Multimer predictions.

*supplement 1D*). Finally, pDockQ and visual analysis of the predictions for the 8-protein complex indicate that SOF1 is mainly disordered and does not make significant contacts with other components (*Figure 5A* and *Figure 5—figure supplement 1A*). Taken together, these considerations leave egg JUNO and CD9 and sperm IZUMO1, SPACA6 and TMEM81 as subunits of a 5-protein complex that can be consistently modeled with acceptable ranking confidence and pDockQ scores (*Figure 5B and C*). Consistent with their central role in interfacing the egg and sperm plasma membranes, JUNO and IZUMO1 constitute the core of this putative assembly, where they interact in the same way that was observed crystallographically (*Aydin et al., 2016*; *Ohto et al., 2016*) and reproduced computationally (*Figure 2*). On the opposite side of the JUNO/IZUMO1 interface, the hinge region and 4HB of SPACA6 wrap around the 4HB of IZUMO1, generating a concave surface that interacts with the long extracellular loop (LEL) of CD9. Finally, TMEM81 adopts the same N-to-C orientation of IZUMO1 and SPACA6 and, by inserting its Ig-like module between the two proteins, links their C-terminal regions.

## Discussion

In this study, we have taken advantage of the latest developments in protein structure and interaction prediction to model protein complex formation in the mammalian egg-sperm fusion synapse. We report the supramolecular organization of five cell surface proteins (three sperm and two egg) that form a core complex likely to be important for gamete recognition and fusion.

Because the only specific information about the target molecules that is used as input for AlphaFold-Multimer is their primary sequence, the neural network model does not incorporate any knowledge of data associated with the system's biology. As a result, biological information on egg-sperm fusion can be used as an independent criterion to validate the predictions. Firstly, because the majority of proteins involved in this process are either C-terminally membrane-anchored or transmembrane proteins, a basic feature expected in a gamete fusion synapse is that the C-termini of its egg subunits or the egg subunits themselves should all be located on the opposite side of the corresponding elements from sperm, relative to the gamete interface. This is true for the 5-component complex predictions (*Figure 5C*). On the egg plasma membrane side, JUNO and CD9 are positioned so that the GPI anchor attached to the C-terminus of the former (which is not modeled by AlphaFold, whose predictions are currently restricted to amino acids) would be located in correspondence with the transmembrane domains of CD9. Similarly, the general orientation and high flexibility of the juxtamembrane regions of IZUMO1, SPACA6, and TMEM81 are compatible with the fact that, in the context of the full-length proteins, these elements are connected to the single-spanning transmembrane helices that anchor the corresponding molecules to the sperm plasma membrane.

A second feature common to all the subunits in the modeled complexes is the presence of N-glycosylation sites. Because AlphaFold has no explicit knowledge of sequons and does not model carbohydrates, the N-glycans that decorate the native molecules could, in principle, interfere with predicted interfaces. As shown in *Figure 6*, the predicted complex architecture is compatible with the location of all the possible N-glycosylation sites of JUNO, IZUMO1, SPACA6, and TMEM81, for both human and mouse homologs (amounting to a total of 10 sites). One possible exception is a sequon within the short extracellular loop (SEL) of CD9, which is conserved in both species (corresponding to human N52 and mouse N50, respectively) but whose glycosylation remains to be experimentally verified. Interestingly, this site is located in relatively close proximity to where loop 3 of JUNO protrudes towards the region between the LEL and SEL of CD9 (*Figure 7A*). This suggests that if the conserved

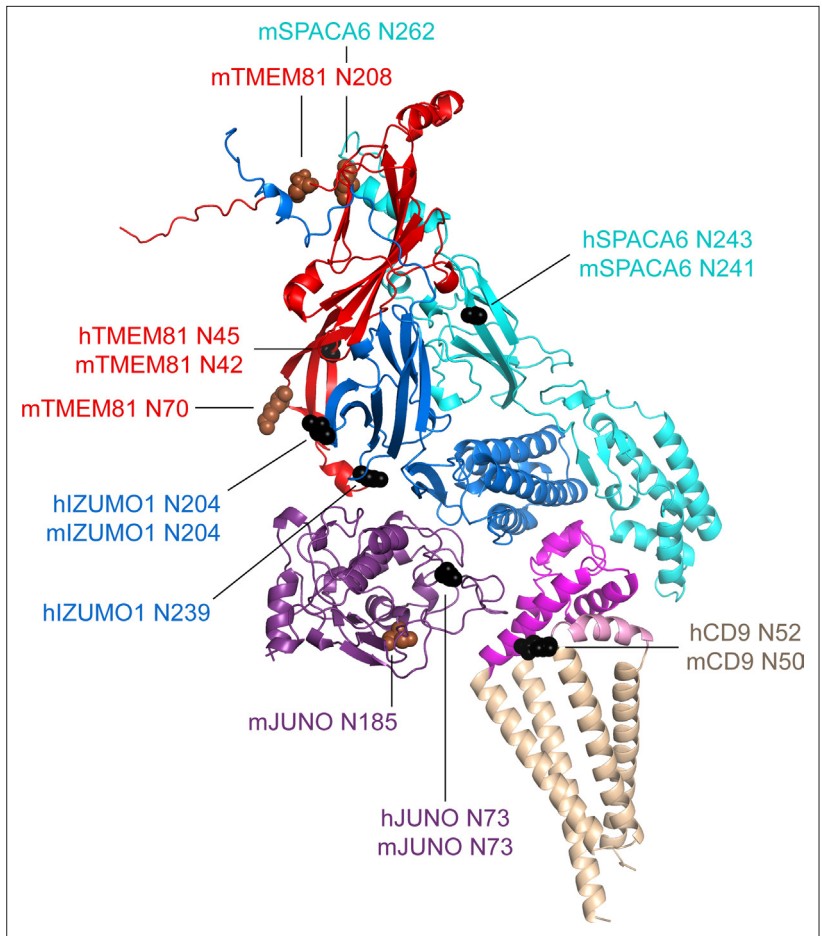

**Figure 6.** Mapping of sequon positions onto the 5-subunit complex prediction. The positions of possible glycosylation sites are mapped onto the model of the 5-subunit assembly (depicted as in **Figure 5C**) by showing the corresponding Asn residues in sphere representation. Sequons found in human (h prefix) or in both human and mouse proteins are colored black, whereas sequons only found in mouse (m prefix) proteins are brown.

The online version of this article includes the following source data for figure 6:

**Source data 1.** Input and output files for the displayed AlphaFold-Multimer prediction.

sequon of CD9 is glycosylated, this may interfere with the only minor contact that the protein makes with JUNO within the predicted complex. Notably, a fusion synapse architecture where CD9 makes little or no contact with JUNO but interacts with the 4HB of IZUMO would immediately explain the experimental observation that, in mouse oocytes, CD9 is recruited to the gamete fusion site only upon binding of JUNO to IZUMO1 (**Chalbi et al., 2014**). Moreover, the predicted CD9/IZUMO1 interface agrees with previous suggestions that the two proteins may interact, based on the observation that their sequences co-evolve (**Claw et al., 2014**; **Vicens and Roldan, 2014**).

Although there is a general agreement that the CD9 LEL plays an important role in gamete fusion, which of its residues are responsible for this is debated; in particular, an early suggestion that the 173-SFQ-175 motif of mouse CD9 LEL is required for fusion was recently challenged (**Umeda et al., 2020**; **Zhu et al., 2002**). Against this background, it is interesting to note that, in our predictions, the conserved CD9 Phe at the center of the SFQ tripeptide (175-TFT-177 in human) stacks against α-helix 2 of the IZUMO 4HB (**Figure 7B**). Not far from this interaction, the third α-helix of CD9 LEL makes hydrophobic contacts with IZUMO1 α2 and α4. These interactions are close to L115, a conserved IZUMO1 α4 residue thought to contribute to egg binding and fusion (**Inoue et al., 2013**), and directly involve W113, another conserved α4 amino acid that was recently implicated in fusion (**Brukman et al., 2023**). Notably, W113 bridges CD9 and SPACA6 by inserting between their LEL and double CXXC motif elements, respectively, while W88 — another IZUMO1 residue suggested to be important

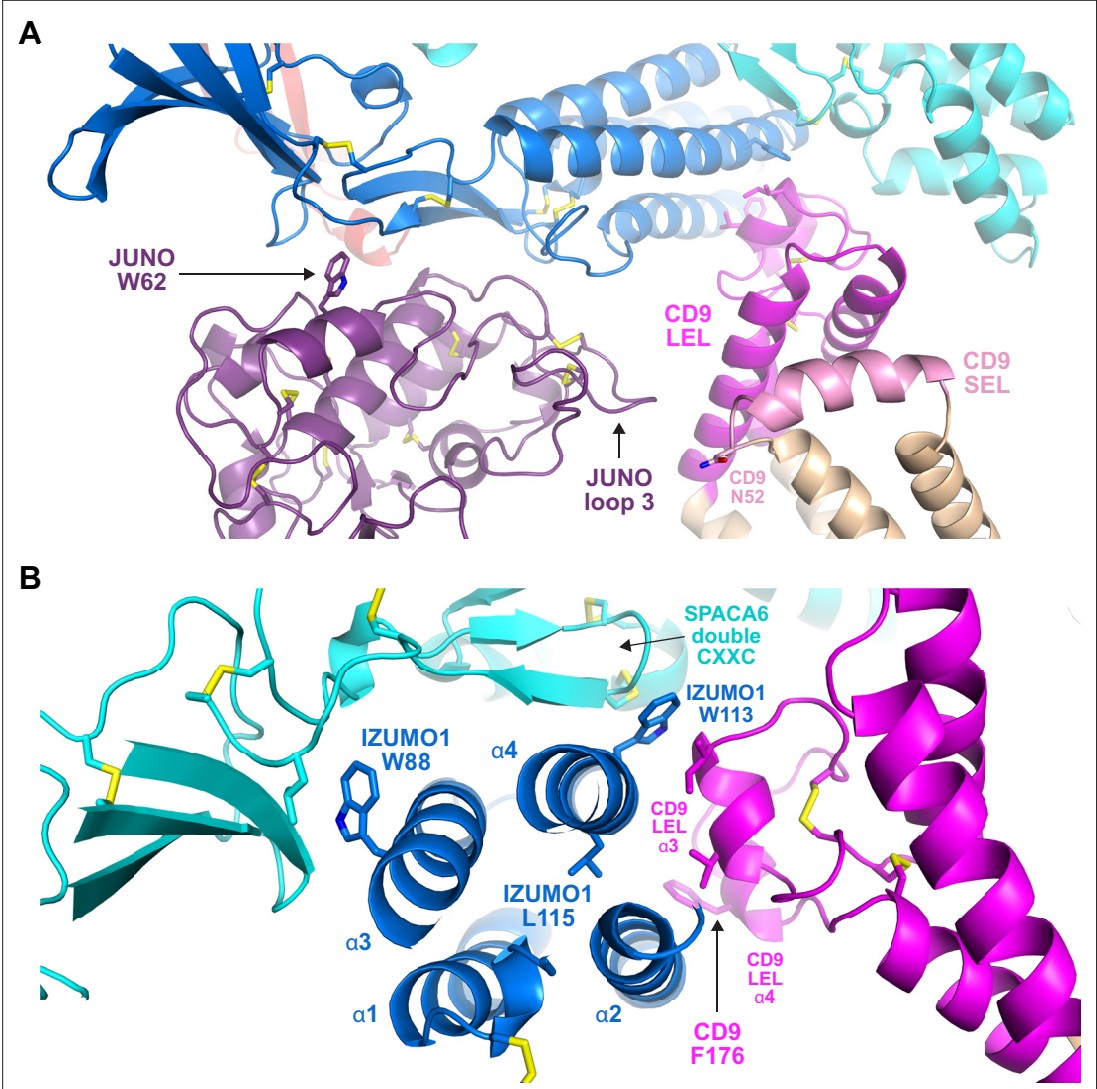

**Figure 7.** Subunit interfaces of the predicted complex involve protein elements previously implicated in fusion. (**A**) Detail of the prediction shown in *Figure 5C*, highlighting functionally important regions of JUNO and CD9, as well putatively N-glycosylated CD9 N52. (**B**) Different view of the same complex prediction, centered around the IZUMO1 4HB.

The online version of this article includes the following source data for figure 7:

**Source data 1.** Input and output files for the displayed AlphaFold-Multimer prediction.

for fusion (*Brukman et al., 2023*) — also interacts hydrophobically with SPACA6 at the opposite side of IZUMO1's 4HB (*Figure 7B*).

Whereas all the data above is in good agreement with the structural predictions described in this manuscript, two aspects should be considered with caution. First, it remains unclear why, despite the fact that IZUMO1 complementation rescues the disappearance of SPACA6 from the mature sperm of IZUMO1 null mice (*Inoue et al., 2021*), attempts to biochemically identify a complex between IZUMO1 and SPACA6 have been met with limited success (*Noda et al., 2020*; *Vance et al., 2022*). Based on the predicted complex architecture, one obvious possibility would be that, in order to be stable, the interaction also requires the presence of TMEM81. One reason could be the low affinity of the interactions between these proteins, which is typical of extracellular receptor-ligand interactions and makes them difficult to detect experimentally (*Wright and Bianchi, 2016*). Also, to avoid any inappropriate membrane fusion events, the individual components of the complex may be purposefully spatially segregated until brought together at the moment of fusion, again making it difficult to detect this complex in vivo. The type of structural modeling approach described here could play a

role in understanding the function of dynamic and transiently formed protein complexes in a range of biological processes that would otherwise be difficult to identify, although care should be taken as some complexes might be predicted due to interactions between homologs. Second, because it is difficult to assess the confidence of the interface between CD9 and IZUMO1+SPACA6 due to its relatively limited extent (combined interface area ~730 $\text{Å}^2$), it cannot be excluded that some protein other than CD9 may be the true counterpart of IZUMO1+SPACA6. In other words, it is, in principle, also possible that AlphaFold-Multimer simply recognizes that the concave surface generated by the combined 4HBs of IZUMO1 and SPACA6 is likely to engage in additional interactions and thus tries to fill it with another suitable input subunit.

Of direct relevance to these questions is an independent study, submitted back to back with the original preprint of the present manuscript, which provides experimental data for the existence of a trimeric IZUMO1/SPACA6/TMEM81 complex in zebrafish and suggests that this interacts with egg Bouncer (*Deneke et al., 2023*). Considering that the mammalian orthologue of Bouncer is expressed on sperm instead of the egg (*Fujihara et al., 2021*) and that role of CD9 in egg-sperm fusion is much more important in mammals than in fish (*Greaves et al., 2022*), the combination of our studies raises the intriguing possibility that, during the course of evolution, CD9 may have substituted Bouncer as a binding partner of the IZUMO1/SPACA6/TMEM81 complex.

## Methods

### Key resources table

| Reagent type (species) or resource | Designation | Source or reference | Identifiers | Additional information |
|---|---|---|---|---|
| Cell line (*Homo sapiens*) | HEK293T | Dr. Radu Aricescu and Dr. Yuguang Zhao (University of Oxford, UK) | | |
| Cell line (*Homo sapiens*) | HEK293S GnTI- | ATCC | CRL-3022; RRID:CVCL_A785 | |
| Antibody | Mouse IgG1 monoclonal Penta·His | QIAGEN | 34660; RRID:AB_2619735 | (1:1000) |
| Antibody | Mouse IgG1 monoclonal Anti-c-Myc (clone 9E10) | Sigma-Aldrich | M4439; RRID:AB_439694 | (1:5000) |
| Antibody | Peroxidase AffiniPure Goat Anti-Mouse IgG (H+L) | Jackson ImmunoResearch Laboratories | 115-035-003; RRID:AB_10015289 | (1:10000) |
| Transfected construct (*Homo sapiens*) | pHLsec3-hJUNO-(GGGS)$_2$H$_8$ | This publication | | Methods section "DNA constructs" |
| Transfected construct (*Homo sapiens*) | pHLsec3-hIZUMO1-Myc | This publication | | Methods section "DNA constructs" |
| Transfected construct (*Homo sapiens*) | pHLsec3-mJuno-H$_8$ | *Han et al., 2016* | | |
| Transfected construct (*Homo sapiens*) | pHLsec3-mIzumo1-LEH$_6$ | *Nishimura et al., 2016* | | |
| Chemical compound, drug | 25 kDa branched polyethyleneimine | Sigma-Aldrich | 408727 | |
| Chemical compound, drug | SimplyBlue SafeStain | Thermo Fisher Scientific | LC6060 | |
| Commercial assay or kit | PCR Mycoplasma Test Kit II | Applichem | A8994 | |
| Software, algorithm | AlphaFold2 | *Jumper et al., 2021*; *Evans et al., 2021* | | |
| Software, algorithm | Belvu | *Barson and Griffiths, 2016* | | |
| Software, algorithm | Dali | *Holm, 2020* | | |

*Continued*

| Reagent type (species) or resource | Designation | Source or reference | Identifiers | Additional information |
|---|---|---|---|---|
| Software, algorithm | Foldseek | *van Kempen et al., 2024* | | |
| Software, algorithm | pDockQ | *Bryant et al., 2022* | | |
| Software, algorithm | PyMOL | Schrödinger, LLC | | |
| Software, algorithm | UCSF Chimera | *Meng et al., 2006* | | |

## DNA constructs

For expression of human JUNO, a synthetic gene encoding the protein's ectodomain (residues G20-S228) followed by a 2 x GGGS linker sequence (ATUM, Newark, CA, USA) was cloned into the *Age*I and *Xho*I restriction sites of mammalian expression vector pHLsec3 (*Raj et al., 2017*), in frame with 5' and 3' sequences encoding a CRYPα signal peptide/ETG tripeptide and an 8His-tag, respectively (construct pHLsec3-hJUNO-(GGGS)$_2$H$_8$). pHLsec3 was also used to express a C-terminally Myc-tagged version of the ectodomain of human IZUMO1, preceded by its signal peptide (residues M1-L283) (construct pHLsec3-hIZUMO1-Myc). The ectodomains of mouse JUNO and IZUMO1 were expressed using previously described constructs pHLsec3-mJuno-H$_8$ and pHLsec3-mIzumo1-LEH$_6$(*Han et al., 2016*; *Nishimura et al., 2016*).

## Cell lines

For transfection experiments, we used two different kinds of HEK293 cells: HEK293T (kindly donated by Dr. Radu Aricescu and Dr. Yuguang Zhao (University of Oxford, UK)) and HEK293S GnTI- (ATCC). Cell line authentication was performed by the respective sources, who also tested for mycoplasma contamination. We independently confirmed that both cell lines were mycoplasma-free using a PCR Mycoplasma Test Kit II (Applichem).

## Protein expression, purification, and analysis

Polyethyleneimine-mediated transient transfection of HEK293 cells and protein purification by immobilized metal affinity chromatography (IMAC) and size-exclusion chromatography (SEC) was carried out following published protocols (*Bokhove et al., 2016*). While human JUNO$_E$-His and IZUMO1$_E$-Myc were always co-transfected, both individual transfection and co-transfection experiments were carried out in the case of mouse JUNO$_E$-His and IZUMO1$_E$-His. Samples separated on SDS-PAGE gels were detected with SimplyBlue SafeStain (Thermo Fisher Scientific) or subjected to immunoblotting with either Penta·His (1:1,000; QIAGEN) or Anti-c-Myc (1:5,000; Sigma-Aldrich) mouse monoclonal antibodies. Secondary antibody was horseradish peroxidase-conjugated goat anti-mouse IgG (1:10,000; Jackson ImmunoResearch Laboratories). Experiments were independently performed as biological replicates 2–3 times, with the same outcome.

## AlphaFold predictions

Predictions were generated with local copies of AlphaFold2 (*Jumper et al., 2021*; *Evans et al., 2021*), installed using versions 2.2–2.3.2 of the open-source code available at (https://github.com/deepmind/alphafold; *google-deepmind, 2024*) or by taking advantage of the Berzelius supercomputing resource (National Supercomputer Centre, Linköping University). All runs were performed using the full_dbs preset and excluding PDB templates. The human protein regions used for the binary interaction predictions whose network is shown in *Figure 4* were CD9 P2-V228 (UniProt P21926); CD81 M1-Y236 (P60033); DCST1 M1-G706 (Q5T197); DCST2 M1-K773 (Q5T1A1); FIMP A22-S77 (Q96LL3-2); JUNO G20-S228 (A6ND01); IZUMO1 C22-Q284 (Q8IYV9); IZUMO2 C21-P183 (Q6UXV1); IZUMO3 C21-D166 (Q5VZ72); IZUMO4 C18-H232 (Q1ZYL8); MAIA Q16-L580 (Q96P31); SOF1 S29-H122 (Q96LL11); SPACA6 C27-T291 (W5XKT8); TMEM81 I31-P218 (Q6P7N7); TMEM95 C17-D140 (Q3KNT9). Additional prediction runs were performed using full-length sequences (excluding N-terminal signal peptide regions) also for sperm type I-transmembrane proteins and sequences that lacked disordered

protein regions. The pDockQ confidence score of multi-chain predictions was calculated as described earlier (*Bryant et al., 2022*).

## Structure analysis and comparison

Model coordinates were visualized, inspected and superimposed with PyMOL (Schrödinger, LLC), which was also used to generate all structural figures. Database searches were carried out using Dali (*Holm, 2020*) and Foldseek (*van Kempen et al., 2024*); structure-based alignments were generated with UCSF Chimera (*Meng et al., 2006*) and manually edited with Belvu (*Barson and Griffiths, 2016*).

## Materials availability

Mammalian expression constructs are available from L.J. upon request.

## Acknowledgements

Computations and data handling were enabled by the supercomputing resource Berzelius provided by the National Supercomputer Centre at Linköping University, the Knut and Alice Wallenberg Foundation, and SNIC (grants Berzelius-2021–29 and SNIC 2021/5-297). We thank Andrea Pauli (IMP, Vienna) for sharing a preprint before submission to bioRxiv. Funding: Swedish Research Council (2020-04936, 2021–03979) Luca Jovine, Arne Elofsson. Knut and Alice Wallenberg Foundation (2018.0042, 2022.0032) Luca Jovine, Arne Elofsson. Biotechnology and Biological Sciences Research Council (BB/T006390/1) Enrica Bianchi, Gavin J Wright.

## Additional information

### Funding

| Funder | Grant reference number | Author |
|---|---|---|
| Knut och Alice Wallenbergs Stiftelse | 2018.0042 | Luca Jovine |
| Vetenskapsrådet | 2020-04936 | Luca Jovine |
| Vetenskapsrådet | 2021-03979 | Arne Elofsson |
| Biotechnology and Biological Sciences Research Council | BB/T006390/1 | Gavin J Wright Enrica Bianchi |
| National Academic Infrastructure for Supercomputing in Sweden | Berzelius-2021–29 | Arne Elofsson |
| Knut och Alice Wallenbergs Stiftelse | 2022.0032 | Arne Elofsson |

The funders had no role in study design, data collection and interpretation, or the decision to submit the work for publication.

### Author contributions

Arne Elofsson, Resources, Software, Formal analysis, Funding acquisition, Investigation, Visualization, Methodology, Writing - review and editing; Ling Han, Validation, Investigation, Methodology; Enrica Bianchi, Gavin J Wright, Funding acquisition, Writing - review and editing; Luca Jovine, Conceptualization, Resources, Formal analysis, Supervision, Funding acquisition, Validation, Investigation, Visualization, Writing - original draft, Project administration, Writing - review and editing

### Author ORCIDs

Arne Elofsson ⓘ http://orcid.org/0000-0002-7115-9751
Enrica Bianchi ⓘ http://orcid.org/0000-0001-8124-7328
Gavin J Wright ⓘ http://orcid.org/0000-0003-0537-0863
Luca Jovine ⓘ http://orcid.org/0000-0002-2679-6946

Reviewer #2 (Public Review): https://doi.org/10.7554/eLife.93131.3.sa1
Reviewer #3 (Public Review): https://doi.org/10.7554/eLife.93131.3.sa2
Author response https://doi.org/10.7554/eLife.93131.3.sa3

## Additional files

### Supplementary files
- Supplementary file 1. Evaluation metrics for protein complex predictions.
- MDAR checklist

### Data availability

Uncropped gel and blot scans are available as Figure 1-Source Data and Figure 1-Figure Supplement 1-Source Data, respectively. Evaluation metrics for protein complex predictions are available as Supplementary File 1, and prediction data used for Figures 2, 4-7 and Figure 5-Figure Supplement 1 has been included in the corresponding Source Data files.

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
