## [Editor Report · eLife assessment]

This study offers **valuable** insights into the structural architecture of the mammalian egg-sperm fusion synapse, shedding light on the role of specific proteins in fertilization. The significance of the findings lies in the potential identification of a pentameric complex involved in gamete fusion by AlphaFold Multimer. The strength of evidence for the approach/methodology is **solid**, while the experimental validation is **incomplete** in supporting these interactions. This work will be of interest to biomedical researchers working on fertility and reproductive health.

---

## [Referee Report · Reviewer #2 (Public Review)]

Summary:

Fertilization is a crucial event in sexual reproduction, but the molecular mechanisms underlying egg-sperm fusion remain elusive. Elofsson A et al. used AlphaFold to explore possible synapse-like assemblies between sperm and egg membrane proteins during fertilization. Using a systematic search of protein-protein interactions, the authors proposed a pentameric complex of three sperm (IZUMO1, SPACA6, and TMEM81) and two egg (JUNO and CD9) proteins, providing a new structural model to be used in future structure-function studies.

Strengths:

(1) The study uses the AlphaFold algorithm to predict higher-order assemblies. This approach could offer insights into a highly transient protein complex, which are challenging to detect experimentally.

(2) The article predicts a pentameric complex between proteins involved in fertilization, shedding light on the architectural aspects of the egg-sperm fusion synapse.

Weaknesses:

The proposed model, which is a prediction from a modeling algorithm, lacks experimental validation of the identity of the components and the predicted contacts.

It is noteworthy that in an independent study, Deneke et al. provides experimental evidence of the interaction between IZUMO1/SPACA6/TMEM81 in zebrafish. This is an important element that supports the findings presented in this manuscript

Regarding the authors response on the question of a global search:

I understand that a global search might be difficult to interpret because a large number of putative false positives. But it is this type of information that is needed to assess the validity of the model and the scoring power in the absence of any experimental validation. At minimum, the search should include a negative control set of proteins known to be unrelated to sperm fertilization or homologous egg-sperm fusion complexes from incompatible species to account for species-specific interactions.

I acknowledge that experimentally validating highly transient complexes presents technical hurdles. However, a high-confidence structural model could enable the design of point mutations specifically disrupting the predicted interactions. Subsequent rescue experiments could then validate the directionality of these interactions. Ultimately, such experiments are crucial for robust model validation.

---

## [Referee Report · Reviewer #3 (Public Review)]

Summary:

Sperm-egg fusion is a critical step in successful fertilization. Although several proteins have been identified in mammals that are required for sperm-egg adhesion and fusion, it is still unclear whether there are other proteins involved in this process and how the reported proteins complex co-operate to complete the fusion process. In this study, the authors first identified TMEM81 as a structural homologue of IZUMO1 and SPACA6, and predicted the interactions with a pool of human proteins associated with gamete fusion, using AlphaFold-Multimer, a recent advance in protein complex structure prediction. The prediction is compelling and well discussed, and the experimental evidence to verify this interaction is lacking in this study but supported by a complementary and independent study by another group.

Strengths:

The authors present a pentameric complex formation of four previously reported proteins involved in egg/sperm interaction together with TMEM181 using a deep learning tool, AlphaFold-Multimer.

Weaknesses:

It is intriguing to see that some of the proteins involved in sperm-egg interaction are successfully predicted to be assembled into a single multimeric structure by AlphaFold-Multimer. The experimental validation of the interactions is not directly supported in this study. As there are more candidate proteins in the process, testing other possible protein interactions more comprehensively will provide more rationale for the current 3D multi-protein modeling.

---

## [Author Response]

The following is the authors’ response to the original reviews.

**Reviewer #1**
The authors should include experiments such as Cryo-EM and genetically modified animals to demonstrate the physiological importance of the TMEM81 complex.

While we intend to pursue cryo-EM studies of the putative complex (or subcomplexes thereof), this is clearly not a straightforward endeavor and goes beyond the scope of the present manuscript. Concerning the generation of genetically modified animals, we would like to underline that the majority of the proteins that we used for AlphaFold-Multimer complex predictions were precisely chosen based on the fact that - as detailed in the publications referenced in the Introduction - ablation of the respective genes caused sex-specific infertility due to defects in gamete fusion (the other criterion used for inclusion being structural similarity to IZUMO1 coupled with expression in the testis (IZUMO2-4 and TMEM81), or evidence from other kinds of experiments in the case of human-specific MAIA). Concerning TMEM81, experimental evidence for a direct involvement in gamete fusion is described in the referenced preprint by Daneke et al., which was submitted to bioRxiv concomitantly with the present work.

**Reviewer #2**
I believe that the manuscript would benefit from the authors providing more information about the systematic search (Figure 4). For example, by indicating for each pair tested the average pDock score in a 2D plot (or table) and as raw data in the supplementary information.

Figure 4 has been modified to report both the top and the mean ranking scores for every interaction. Furthermore, additional metrics for the systematic search summarized in Figure 4, including pDockQ scores, are provided in this manuscript revision as supplementary Table S1.

A global search, such as including all membrane proteins expressed in eggs or sperm, could not only be more informative but could also allow the reader to understand the pDock score discrimination power for this particular subset.

The possibility of carrying out a global search was evaluated by performing preliminary computational experiments on an extended ensemble of sperm and egg proteins. In order to do so, we compiled a list of sperm membrane proteins by referring to 4 proteomic datasets (PMIDs 36384108, 36896575, 31824947, 24082039) and identifying ~600 proteins that were found in at least two of them; among these, 250 were single-pass type I or type II membrane proteins, or GPI-anchored proteins. Similarly, a list of 160 egg surface membrane proteins, excluding multipass and secreted ones, was obtained by comparing oocyte cDNA library NIH_MGC_257_N (Express Genomics, USA) with 4 proteomic datasets (PMIDs 35809850, 36042231, 29025019, 27215607). As we briefly commented at the beginning of the section “Prediction of interactions between human proteins associated with gamete fusion” of the revised manuscript, the tests carried out using the resulting list of sperm and egg proteins suggested that interpreting the results of a global search would be severely complicated by a relatively large number of putative false positives. Moreover, the tests showed that performing a complete systematic search would be beyond our current access to computing power. Based on these observations, we preferred to maintain the present study limited to proteins that had been previously clearly implicated in gamete fusion and/or matched specific structural features of IZUMO1.

Figure 5 could be improved in clarity by schematically indicating to which cell each protein is anchored.

This has been done in the revised version of the manuscript.

**Reviewer #3**
Major comments(1) In Figure 1, how the protein of mouse/human IZUMO1 and JUNO is purified is not mentioned in the main text nor in the Methods. Are the mouse IZUMO1-His and mouse JUNO-His transfected together or separately? Are human JUNO-His and human IZUMO1-Myc transfected together into HEK293 cells? And purified by IMAC?

Transfection information has been included in the Methods section “Protein expression, purification and analysis” (previously “Protein expression and purification”). Concerning the purification procedure, we had already stated in the legend of Figure 1 that human JUNOE-His/IZUMO1E-Myc had been purified by IMAC before SEC, and have now done the same for mouse JUNOE-His and IZUMO1E-His.

(2) It would be easier to understand the figure if the author could run a WB to indicate which band above JUNO is specifically IZUMO1-Myc in Figure 1.

This has been done and reported in a new Figure S1 (with the original Figure S1 having now become Figure S2). Details about the antibodies used for immunoblot have been included in both Methods section “Protein expression, purification and analysis” and the Key Resources Table.

(3) Figure 4: Analysis of more proteins that have been suggested as possible candidates for sperm-egg interaction will help to highlight the following results. Also, providing a score for the possibility of interaction might help in selecting those proteins in Figures 5 and 6.

Please refer to the answer to the first question of Reviewer #2.

(4) Figure 7: The authors take advantage of the latest developments in protein structure and interaction to model protein complex formation. However, some experimental experiments such as Co-IP, pull down to support the prediction to verify some of this predicated interaction is necessary.

We agree with the reviewer; however, for the reasons we discussed during our comparison of the biochemical properties of the JUNO/IZUMO1 interaction between mouse and human, pursuing this line of inquiry will likely necessitate an extensive set of parallel experiments using proteins from different species. This work is being planned and will be the focus of future studies. However, as we mentioned at the end of the Abstract, one should also consider that some of these complexes are likely to be highly transient. Because of this, while they may have important regulated roles in vivo (function at a specific time and place), they could be very challenging to detect using standard approaches in vitro. We thus see this as a significant advance that structural modeling could contribute to the identification of such functionally important but transient interactions.

Minor points

(1) In the abstract, "three sperm (IZUMO1, SPACA6 and TMEM81) "should be "three sperm proteins."

The Abstract has been condensed to fit within the suggested 200-word limit and, as part of this, the sentence has been changed to “complex involving sperm IZUMO1, SPACA6, TMEM81 and egg JUNO, CD9”.

(2) How do the predictions of the binary complex IZUMO1/CD9 (Figure S1B) or IZUMO1/CD81 (Figure S1C) suggest "the two egg tetraspanins are interchangeable"? Was it because they are quite similar? Please provide more explanation for this speculation. Interchangeable by function or for complex formation? To support the conclusion, biochemical data is required. Otherwise, it needs to be toned down.

This is because, in the AlphaFold-Multimer predictions of the pentameric complex, CD9 and CD81 are placed in essentially the same way relative to the other subunits.

We have now clarified this at the end of page 6:

“(...) suggest that the two egg tetraspanins are interchangeable because they are predicted to bind to the same region of IZUMO1; (...)”

(3) It would be more reader-friendly if the author could label the name of each protein in the figure in Figure S1, especially when the name is not written in the figure legend.

This has been done in Figure S2 of the revised manuscript (corresponding to original Figure S1).